# Measurement of symphysis fundal height for gestational age estimation in low-to-middle-income countries: A systematic review and meta-analysis

**Rachel Whelan[1‡], Lauren Schaeffer[1‡], Ingrid Olson[1], Lian V. Folger[1,2], Saima Alam[3], Nayab Ajaz[4], Karima Ladhani[5], Bernard Rosner[6], Blair J. Wylie[7,8‡], Anne C. C. Lee[1,8‡]***

1 Global Advancement of Infants and Mothers (AIM) Lab, Department of Pediatric Newborn Medicine, Brigham and Women's Hospital, Boston, MA, United States of America, 2 Department of Maternal and Child Health, University of North Carolina Chapel, Hill Gillings School of Global Public Health, Chapel Hill, NC, United States of America, 3 Berkshire Medical Center, Pittsfield, MA, United States of America, 4 Tufts University School of Medicine, Boston, MA, United States of America, 5 Department of Global Health and Population, Harvard T.H. Chan School of Public Health, Boston, MA, United States of America, 6 Department of Biostatistics, Harvard T.H. Chan School of Public Health, Boston, MA, United States of America, 7 Division of Maternal-Fetal Medicine, Department of Obstetrics and Gynecology, Beth Israel Deaconess Medical Center, Boston, MA, United States of America, 8 Harvard Medical School, Boston, MA, United States of America

‡ RW and LS share first authorship on this work. BJW and ACCL are joint senior authors on this work.
* alee6@bwh.harvard.edu

**Data Availability Statement:** All data is fully available without restriction. No original data were generated from this systematic review. All

## Abstract

In low- and middle-income countries (LMIC), measurement of symphysis fundal height (SFH) is often the only available method of estimating gestational age (GA) in pregnancy. This systematic review aims to summarize methods of SFH measurement and assess the accuracy of SFH for the purpose of GA estimation. We searched PubMed, EMBASE, Cochrane, Web of Science, POPLINE, and WHO Global Health Libraries from January 1980 through November 2021. For SFH accuracy, we pooled the variance of the mean difference between GA confirmed by ultrasound versus SFH. Of 1,003 studies identified, 37 studies were included. Nineteen different SFH measurement techniques and 13 SFH-to-GA conversion methods were identified. In pooled analysis of five studies (n = 5838 pregnancies), 71% (95% CI: 66–77%) of pregnancies dated by SFH were within ±14 days of ultrasound confirmed dating. Using the 1 cm SFH = 1wk assumption, SFH underestimated GA compared with ultrasound-confirmed GA (mean bias: -14.0 days) with poor accuracy (95% limits of agreement [LOA]: ±42.8 days; n = 3 studies, 2447 pregnancies). Statistical modeling of three serial SFH measurements performed better, but accuracy was still poor (95% LOA ±33 days; n = 4 studies, 4391 pregnancies). In conclusion, there is wide variation in SFH measurement and SFH-to-GA conversion techniques. SFH is inaccurate for estimating GA and should not be used for GA dating. Increasing access to quality ultrasonography early in pregnancy should be prioritized to improve gestational age assessment in LMIC.

extracted data is available within the existing tables and Supporting Information.

**Funding:** This work was supported by the Bill & Melinda Gates Foundation through grant OPP1130198. This work was conducted with support from Harvard Catalyst | The Harvard Clinical and Translational Science Center (National Center for Advancing Translational Sciences, National Institutes of Health Award UL 1TR002541) and financial contributions from Harvard University and its affiliated academic healthcare centers. ACL was supported by a grant from the Eunice Kennedy Shriver National Institute of Health and Child Development (K23 HD091390-01). The content is solely the responsibility of the authors and does not necessarily represent the official views of Harvard Catalyst, Harvard University and its affiliated academic healthcare centers, or the National Institutes of Health. The funders had no role in study design, data collection and analysis, decision to publish, or preparation of the manuscript.

**Competing interests:** This research was supported by a grant from the Bill and Melinda Gates Foundation (BMGF). ACL reported research grants from the Eunice Kennedy Shriver National Institute of Health and Child Development and the BMGF, and is a consultant to the World Health Organization. BW and BR reported research grants from the NIH. BW has served on the Board for the Society of Maternal-Fetal Medicine within the past three years. This does not alter our adherence to PLOS One policies on sharing data and materials.

# Introduction

Globally, measurement of symphysis-fundal height (SFH)–the distance from the symphysis pubis to the top of the uterine fundus–is routinely used in clinical practice for monitoring of fetal growth during pregnancy to identify fetuses at higher risk for perinatal morbidity and mortality. International reference standards for SFH at each week of gestation have been developed for healthy fetal growth based on optimally healthy cohorts of pregnant women [1, 2].

While SFH is primarily used for fetal growth monitoring in high-income countries (HIC), SFH is also commonly used in low- and middle-income countries (LMIC) to estimate gestational age (GA), due to lack of access to more accurate dating methods [3]. Accurate pregnancy dating is necessary for clinical decision-making, including targeted administration of life-saving interventions like antenatal corticosteroids to mitigate preterm complications and the identification and triage of preterm infants [4]. Ultrasound in early pregnancy before 20 weeks' gestation has the highest accuracy for gestational age dating, and new sonography parameters and equations for dating in late pregnancy have shown improved accuracy [5–7]. However, access to ultrasound remains sparse in LMIC. Maternal recall of the first day of the last menstrual period (LMP) is another commonly used method to date pregnancies, but its accuracy is limited by variation in menstrual cycle length, misinterpretation of early bleeding, and poor recall.

When ultrasound and reliable LMP are not available, SFH is frequently used for GA estimation because it is a simple, low-cost, and feasible technique that can be performed by lay health workers [3]. However, fundamental flaws in using SFH for this purpose include the underlying assumption that fetal size approximates GA, and that every fetus of a certain size is the same GA. Fetal size is influenced by genetic factors and normal biologic variation, and fetal growth is influenced by maternal nutrition, health, and morbidities, including infections, pregnancy complications, or environmental exposures. Risk factors for poor fetal growth are much more prevalent in LMICs [8–10]. The use of standard SFH curves from high income settings with low prevalence of these risk factors would, thus, tend to systematically underestimate GA when applied to a population with high prevalence of fetal growth restriction. Further confounding the use of SFH is the lack of standardized methods for measurement of SFH and converting the measurement to gestational age.

The 2016 World Health Organization (WHO) ANC Guidelines concluded that there was inadequate evidence on the role of SFH monitoring in antenatal care [11]. Previous systematic reviews have assessed SFH measurement as a tool for fetal growth monitoring [12–15], however, there is limited data on the accuracy of SFH for estimation of GA. One recent systematic review assessed maternal SFH for GA estimation and concluded "ultrasound-based" measures were more accurate [7], though few SFH studies were identified and prediction accuracy was not summarized for SFH. The purpose of this systematic review is to summarize methods of SFH measurement for GA estimation, existing population-based SFH references, and the accuracy of SFH measurement specifically for GA estimation, among general obstetric populations in LMIC, populations representative of those who would be seen in routine clinical practice in these settings.

# Methods

## Search strategy

A systematic review of the published and gray literature from PubMed, Embase, Cochrane, Web of Science, POPLINE, and the WHO Global Health Libraries and regional databases was conducted from January 1980 up to November 2021. The review was registered with the

International Prospective Register of Systematic Reviews (CRD42015020499) and reported according to the Meta-analysis Of Observational Studies in Epidemiology (MOOSE) statement [16] and the Preferred Reporting Items for Systematic Reviews and Meta-Analyses (PRISMA) statement [17]. The detailed search terms are available in S1 Text in S1 Appendix. Articles were also identified from bibliographies of manuscripts of interest. No language restrictions were applied. Abstracts of non-English articles were translated via Google Translate, and if eligible, the full text was translated into English by fluent speakers. For meeting abstracts, attempts were made to contact the corresponding author to obtain the updated, full text of research.

Studies were eligible for inclusion if the study provided any information about a technique for measuring SFH or reported inter- or intra-rater reliability between SFH measurers. We included studies that reported at least one statistic comparing gestational age determined by SFH and another method (ultrasound or last menstrual period) and enrolled a general, unselected obstetric population. Finally, we also included studies that reported population-based SFH measurements (average or median) by week of pregnancy for cohorts in LMICs with ultrasound confirmed dating. This subset of studies was limited to LMIC-based studies as SFH charts are more likely to be used for GA-dating in LMIC settings, while in HIC, pregnancies are typically dated by ultrasound and SFH charts are used for fetal growth monitoring purposes. Countries were classified as LMIC according to the World Bank at the time of the study's publication [18].

Studies were excluded if they enrolled a highly selected or specialized subpopulation that did not represent the general obstetric population (e.g., only HIV-positive mothers, or strict eligibility criteria of optimally healthy populations within narrow BMI thresholds; with the exception of studies that reported on inter- or intra-reliability between SFH measurers), editorials or reviews without original data, individual case reports, and duplicate search results. We also excluded studies that had <50 patients, or reported only SFH accuracy data for growth monitoring (e.g., determining estimated fetal weight or small-for-gestational age). Institutional Review Board approval was not required for this work.

## Data extraction

Two independent researchers reviewed studies and extracted relevant data into a standard Excel file created for the purpose of this review (S2 Text in S1 Appendix). Differences were resolved through discussion between the two reviewers, or by a third independent reviewer. Characteristics of the included studies can be found in S1 Table in S1 Appendix.

## Study quality assessment

For studies that assessed the accuracy of SFH to estimate GA, the risk of bias was graded by two independent reviewers using the Quality Assessment of Diagnostic Accuracy Studies 2 (QUADAS-2) tool [19], which was modified for the nature of this review (S3 Text in S1 Appendix). Each study was evaluated for potential biases across four domains: (i) patient selection, (ii) test method, (iii) reference (i.e., gold) standard, and (iv) patient flow and timing in pregnancy of SFH measurements. Factors that were considered to potentially influence the relationship between SFH and GA were selected for grading of study quality, and also considered for sub-group/sensitivity analysis. Studies with a gold standard GA based on ultrasound or by ultrasound-confirmation of the menstrual dates (hereafter referred to as 'ultrasound-confirmed' GA) were graded as highest quality given that the ultrasound confirmation of dating would be considered as closest to the "actual" truth.

## Statistical analysis

Stata 15 (StataCorp, College Station, TX) was used for analyses. Studies were grouped by WHO world region and GA gold standard; data were summarized by these groupings using simple descriptive statistics. Data presented in the main manuscript are studies with the highest quality ultrasound-confirmed GA; data from studies identified with the lower quality LMP based dating are shown in the webappendix. For population-based SFH reference studies from LMICs, studies were grouped by the WHO world region given *a priori* differences in fetal size and rates of growth restriction between Africa and Asia [20].

To assess data on the accuracy of GA estimated by SFH, we summarized data on the difference in GA determined by SFH compared to ultrasound-confirmed GA (reference gold standard), as well as the distribution or spread of the differences (standard deviation [SD] of the mean difference, 95% limits of agreement in Bland-Altman analysis, or 95% prediction error in statistical models). For studies that reported Bland-Altman limits of agreement (LOA), the standard deviation was calculated by dividing the 95% LOA/(2*1.96). For studies that reported upon prediction intervals from statistical models, we assumed normality and symmetry of the distribution of residuals. The pooled variance and standard deviation were calculated using the following formula:

$$Variance_{pooled} = \frac{\sum_{i=1}^{k}(n_i - 1)s_i^2}{\sum_{i=1}^{k}(n_i - 1)}$$

To pool studies in which investigators reported upon the proportion of test measured within ±1 to 2 weeks of the gold standard, proportions were logit transformed and standard errors calculated with the equation: SE(logit(p)) = SE(p)/(p*(1-p)), where p is the proportion [21]. Meta-analysis was conducted with a random effects model. Heterogeneity was assessed with the Higgins $I^2$ statistic.

Assessment of publication bias is recommended by Cochrane for meta-analyses with ≥10 studies, as fewer studies do not have adequate power to distinguish real asymmetry from chance [22]. We had no analyses meeting this threshold and were unable to assess for publication bias.

## Results

Of 1606 papers identified, 1003 unique studies were screened by title/abstract, and 37 articles were included (Fig 1). Detailed characteristics of included studies can be found in S1 Table in S1 Appendix. The studies were published between January 1983 and May 2020, with 30 studies from LMICs (12 from Africa, 16 from Asia, 1 from South America, 1 multiregional) and 7 from HIC (3 from North America, 1 from Europe, 1 from Asia, 1 from Oceania, 1 multiregional). Nineteen studies had a gold standard of ultrasound-confirmed GA, 18 studies had LMP-based dating. For additional details on individual study methods, see S1 Table in S1 Appendix.

For studies of diagnostic accuracy, study quality was summarized using QUADAS-2 in S1 Fig in S1 Appendix. Half of the studies (n = 9/17, 53%) had a high risk of bias related to the SFH methodology due to limited descriptions of SFH measurement technique or method of calculation of GA from SFH, absence of quality control procedures, and/or lack of blinding to gold standard GA dating. About 40% (n = 7/17) of the included studies had a high risk of bias related to the gold standard methodology because the gold standard was LMP or not well described.

### SFH measurement techniques

We identified 19 different methods of measuring SFH reported in the literature (Table 1), [23–42] most of which were described in a 1993 review article by Engstrom and Sittler [43].

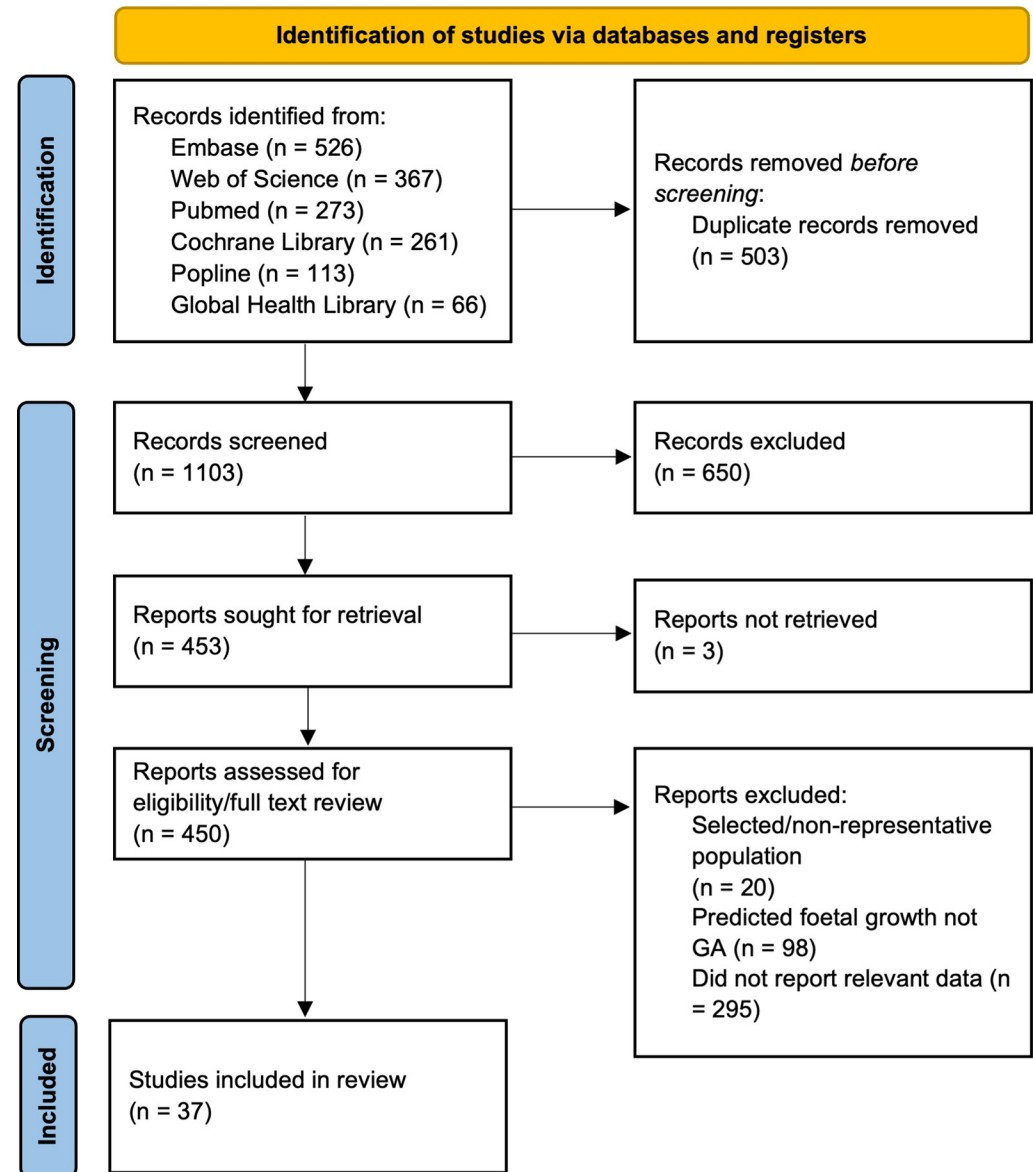

**Fig 1. PRISMA flow diagram for systematic literature review of symphysis fundal height for gestational age estimation.**

Techniques differed by the instrument used to measure SFH (tape measure, caliper, finger-width, ultrasound); choice of superior and inferior landmarks; axis of measurement (vertically at the midline or diagonally at the highest point of the uterine fundus or fetal pole); and by whether the tape was held in contact with the skin of the maternal abdomen or taken straight between two hands.

## SFH to GA conversion methods

Across the studies identified in the review, we identified a wide range of methods by which SFH is used to estimate GA (S2 Table in S1 Appendix). A commonly used clinical rule of thumb is that 1 cm is equal to 1 week for ≥20 weeks' gestation (referred to as 1cm = 1wk in this report) [44, 45]. A similar rule known as McDonald's Rule, first published in 1906,

**Table 1. Symphysis fundal height measurement techniques.**

| Author(s) of Technique<br>*Country* | Description | Instrument | Upper to Lower Anatomic Landmarks |
|---|---|---|---|
| **Tape Measure Technique** | | | |
| **Spiegelberg (1865, 1887)**<br>*Germany* | Midline measurement identifying the margin of the fundus by percussion and measuring the "Length of the Line" connecting the highest point of the fundus with upper edge of symphysis pubis | Tape measure | Highest point of fundus to upper edge of the symphysis pubis |
| **McDonald measurement (1906, 1910)**<br>*USA* | One hand holding the tape in upper border of symphysis pubis and extended fingers of other hand placed perpendicular of uterine fundus then the tape measure grasped with fundal hand and tape is pressed in palm of the hand | Tape measure | Uppermost point of the uterine fundus to upper border of the symphysis pubis |
| **Spalding (1913)**<br>*USA* | Midline measurement with one end of tape in the upper border of symphysis pubis, other in xiphoid process then locating the uppermost point of fundus and read the corresponding number. | Tape measure | Uppermost point of the uterine fundus to upper border of the symphysis pubis |
| **Willson (1958)**<br>*USA* | Extended finger of each hand held perpendicular to uterine fundus and symphysis pubis, and tape measure is held in straight line between two hands. Tape measure not in contact with maternal abdomen at any point. | Tape measure | Uppermost point of the uterine fundus to symphysis pubis |
| **Westin (1977)**<br>*Sweden* | Measured along the longitudinal the axis of uterus, regardless if in the midline. Tape is in contact with maternal abdomen, but not necessarily brought completely to the curve of fundus. Tape is held in end of long axis of uterus | Tape measure | Uppermost point of the uterine fundus to inferior border of symphysis pubis |
| **Belizan (1978)**<br>*Argentina* | Measured from the upper border of the symphysis pubis to the superior fundus uteri, using the cubital edge of the hand to sustain the tape while attempting to reach the middle part of the fundus uteri | Non-elastic tape measure | Superior fundus uteri to upper border of symphysis pubis |
| **Kennedy (1979)**<br>*Botswana* | Measurement blinded to gestational age. Abdomen is divided into quarters above and below the umbilicus to more easily plot the position. | Tape measure | Fundus to symphysis pubis |
| **Garde (1986)**<br>*South Africa* | Bladder must be empty. Upper curve of the fundus is seen by palpating both sides. Highest point is marked on the skin and checked with the index finger parallel to it, pushing backwards. Uterine fundus touches the lateral border of the finder when the mark is correctly placed. Distance is measured along the curve of the skin, without depressing it. | Tape measure | Highest point of fundus to the upper border of the symphysis pubis |
| **Engstrom & Chen (1984;** *USA*), **Linasmita (1984;** *Thailand*); **Varney (1987;** *USA*) | One end of the tape measure on the uppermost border of the symphysis pubis, then identify uppermost border of the uterine fundus and place the ulnar aspect of the other hand perpendicular to the long axis of the uterus. Bring tape measure over fundal hand and record fundal height at point where fundal hand intercepts the tape measure. | Tape measure | Uppermost point of the uterine fundus to upper border (or crest; Engstrom & Chen) of the symphysis pubis |
| **Engstrom (1988)**<br>*USA* | Fingerbreadths are used in place of a tape measure to estimate centimeters above or below the selected landmark, or as a fraction of the distance between two landmarks (e.g. halfway or one-quarter of the way) | No instrument | Not specified |
| **Euans (1995)**<br>*USA* | Palpate the uterine fundus, measurement made from symphysis to fundus, over the fetal axis, with relaxed abdominal and uterine musculature. | Tape measure face down | Symphysis pubis to fundus |
| **Euans (1995)**<br>*USA* | Position the transducer at the superior aspect of the uterus so that the top of the fundus is visible on the imaging screen. Place a finger under the probe until its shadow coincides with the uppermost aspect of the uterus and mark this point on the abdomen. Distance from the superior aspect of the pubic symphysis to this point represents the true fundal height. | Ultrasound | Symphysis pubis to uppermost point of the uterus |
| **Gardosi (1995)**<br>*UK* | Measure starting from the fundus to the symphysis pubis with the tape measure face down. Recommend serial plotting by the same observer. | Tape measure face down | Fundus to symphysis pubis |
| **Caliper Technique—Internal** | | | |

*(Continued)*

**Table 1.** (Continued)

| Author(s) of Technique Country | Description | Instrument | Upper to Lower Anatomic Landmarks |
|---|---|---|---|
| **Ahlfeld: Internal Caliper (1871)** *Germany* | Midline measurement with one branch of caliper is placed in maternal vagina against fetal head and other part of caliper in fetal pole of maternal uterus and obtain measurement of fetal axis | Pelvimetry Caliper | Uppermost point of uterine fundus to vagina against fetal head |
| **Vogt (1922)** *USA* | Index and middle fingers placed into the vagina against the fetal head. Measure the distance between the fetal buttocks in the uterine fundus and a specified point on the examining hand, and subtract the distance between that point on the examining hand and the tip of the fingers from the measurement. | Pelvimetry Caliper | Uterine fundus to fingers placed in vagina against fetal head |
| **Poulos & Langstad (1953)** *USA* | Same as Ahlfeld technique (different lower landmark); recommended to secure caliper with rubber band to examining hand | Pelvimetry Caliper | Uterine fundus to finger on fetal head through maternal rectum |
| **Caliper Technique–External** | | | |
| **Ahlfeld: External Caliper (1871)** *Germany* | One branch of caliper is placed 0.75 cm below the superior border of the symphysis pubis and the other branch of the caliper placed at the uppermost border of the fundus in the midline of the maternal abdomen. | External caliper | Uterine fundus to .75cm below the superior border of symphysis pubis |
| **Reynolds & Baker (1951)** *USA* | Similar to Ahlfeld technique (different lower landmark) | External caliper | Uterine fundus to inferior border of symphysis pubis |
| **Poulos & Langstadt (1953)** | Similar to Ahlfeld technique (different lower landmark) | External caliper | Uterine fundus to superior border of symphysis pubis |

indicates that SFH in cm is equal to GA in weeks for weeks 16 through 32, and then increases by 1 cm every 2 weeks [34]. A "rule of four," referenced in the French Association of Gynecologists and Obstetricians syllabus for residents and commonly used in clinical practice in Rwanda, states that 4 should be added to SFH in cm to estimate GA in weeks (e.g., an SFH of 20 cm is equal to 24 weeks' gestation) [46]. The Global Network developed a color-coded SFH tape measure to classify pregnancies in one of three GA categories: red zone (24–36 weeks), yellow zone (<24 weeks) or green zone (>36 weeks), with different thresholds for Africa vs. Asia [47].

Statistical models have also been developed by many groups and investigators to estimate GA from single or serial SFH measurements (S2 Table in S1 Appendix) [44, 48–54]. White *et al*. developed an online calculator to estimate GA from SFH based on statistical modeling of serial SFH measures from a pregnancy cohort on the Thai-Myanmar border [54].

### Population-based reference curves of the relationship between SFH and GA

We extracted data from studies that reported average or median SFH measurements for general obstetric populations in LMIC across weeks 20 to 40 of gestation. Nineteen unique studies (8 in Africa, 11 in Asia, 1 in South America) reported population-based reference values of SFH in LMIC [44, 47–65]. One study from Panama displayed graphs with 10% and 90% values, however did not display values or report on the 50% or mean and thus was not included [9]. Eight studies had ultrasound confirmed dating as the GA gold standard (Table 2) [44, 47, 50, 53, 54, 56, 61, 65] and 11 used LMP dating (S3 Table in S1 Appendix).

### Accuracy of SFH to estimate GA

Ten studies comparing GA dating by SFH to a high quality gold standard of ultrasound-confirmed GA are shown in Table 3 [44, 45, 47, 53, 54, 66–70]. The range of the timing of SFH

**Table 2. Population-based reference data of SFH measurements (cm) by ultrasound confirmed gestational age (weeks) dates in low-middle income countries.**

| Author & Year | Country & Study Setting | Sample size | Mean/ Median | 20 | 22 | 24 | 26 | 28 | 30 | 32 | 33 | 34 | 35 | 36 | 37 | 38 | 39 | 40 | 41 |
|---|---|---|---|---|---|---|---|---|---|---|---|---|---|---|---|---|---|---|---|
| **African Studies** | | | | | | | | | | | | | | | | | | | |
| Althabe (2015) | Kinshasa, DRC | 671 | Median*♦ | | | 24.3 | 25.8 | 26.8 | 30.0 | 31.0 | 32.4 | 34.0 | 34.5 | 36.0 | | | | | |
| Challis (2002) | Maputo, Mozambique | 817 | Mean* | 19.0 | | 23.0 | | 26.8 | | 30.0 | | | | 33.0 | | | | 35.0 | |
| Kiserud (1986) | Arba Minch, Ethiopia | 114 | Mean (Curve)* †▽ | 18.2 | 19.7 | 22.0 | 23.7 | 25.8 | 27.5 | 29.4 | 30.2 | 31.0 | 31.8 | 32.7 | 33.4 | 33.8 | 34.2 | 34.6 | 34.9 |
| Mador (2011) | Jos, Nigeria | 405 | Mean*†‡♦ | 18.9 | 22.5 | 23.9 | 25.6 | 28.2 | 29.8 | 31.9 | 32.8 | 33.4 | 33.9 | 35.7 | 36.7 | 38.3 | 38.1 | 39.1 | |
| | | | Median♦ | 19.1 | 23.0 | 24.4 | 25.6 | 28.3 | 29.5 | 32.0 | 32.9 | 33.2 | 34.2 | 35.8 | 36.1 | 38.1 | 39.0 | 39.3 | |
| Van Bogaert (1999) | Eastern Cape Province, South Africa | 800 | Mean (Curve)* | 19.7 | 21.4 | 23.4 | 25.2 | 26.8 | 28.5 | 30.4 | 31.1 | 32.1 | 32.9 | 33.9 | 34.7 | 35.4 | 36.2 | 37.1 | |
| **Asian Studies** | | | | | | | | | | | | | | | | | | | |
| Althabe (2015) | Balgaum, India & Karachi, Pakistan | 1089 | Median*♦ | | | 22.5 | 24.0 | 26.0 | 28.0 | 30.0 | 30.5 | 31.3 | 32.5 | 32.3 | | | | | |
| Lee (2020) | Sylhet, Bangladesh | 1146 | Mean*†♦ | 21.8 | 22.8 | 23.2 | 23.4 | 25.1 | 26.9 | 27.8 | 28.0 | 29.3 | 29.5 | 30.7 | 31.3 | 31.6 | 32.1 | 31.6 | 32.4 |
| | | | Median♦ | 21.5 | 22.4 | 23.1 | 23.1 | 24.9 | 26.7 | 27.7 | 28.1 | 29.4 | 29.3 | 30.3 | 31.2 | 31.7 | 32.1 | 31.5 | 32.4 |
| Rao (2014) | Sullia, India | 100 | Mean* | 19.0 | | 23.0 | | 26.8 | | 30.0 | | | | 33.2 | | | | 35.4 | |
| White (2012) | Thailand, Maela refugee camp | 2437 | Mean (Curve)* | 17.4 | 19.1 | 22.6 | 24.0 | 25.8 | 27.2 | 28.7 | 29.0 | 30.2 | 30.3 | 31.5 | 31.9 | 32.1 | 32.8 | 33.3 | 33.5 |

An empty cell indicates that the data was not available for that paper. Sample size was number of pregnant women in the study. Abbreviations: SFH = symphysis fundal height, GA = gestational age, DRC = Democratic Republic of Congo.

\* Indicates inclusion in weighted population-based reference curve created in the current study (with shading in Mean/Median column; only Means from studies with ultrasound as the gold standard reference were included in the weighted population-based reference curve for this study)

† Indicates paper also has standard deviations for each gestational age week listed; ‡ Indicates paper also has standard error for each gestational age week listed

▽ Indicates study excluded preterm infants

♦ Indicates paper also has sample size (number of women) for each gestational age week listed

◇ Indicates population-based reference data study was presented in an inverted table (listed average weeks of gestational age for each whole SFH cm measurement).

\*\* Pelotas, Brazil; Beijing, China; Nagpur, India; Turin, Italy; Nairobi, Kenya; Muscat, Oman; Oxford, UK; Seattle, USA; institutions providing obstetric care with no or low levels of major, known, non-microbiological contamination.

measures in pregnancy for each study is shown in S1 Table in S1 Appendix. Seven studies had an LMP reference are shown in appendix Table S4 in S1 Appendix [48, 52, 71–75].

Three studies reported upon the accuracy of GA estimated using the assumption that 1cm SFH is equal to 1 week of gestation, compared to ultrasound confirmed dating [44, 45, 70]. The precision error in the SFH GA estimate is reflected in the SD of the mean difference. In pooled analysis (n = 2,447 pregnancies), the mean difference between GA estimated by a single SFH measurement (1cm SFH = 1wk GA assumption) and ultrasound-confirmed GA was -14.0 days, with a pooled SD of 21.4 days (95% CI of ±42.8 days). This negative bias towards under-estimation of GA was likely strongly influenced by one study in Bangladesh, in which rates of SGA were high [44]. In sensitivity analysis excluding this study, the pooled mean bias was +1 day and was not significant.

We then assessed the accuracy of studies with a ultrasound-confirmed GA reference that used statistical models of SFH measures to predict GA. In the two studies (n = 1834 pregnancies) [44, 66] that used single measures of SFH to estimate GA, the mean bias was 0.0 and the precision error, or pooled SD, was 22.8 days, indicating that the statistical models predicted 95% of GA estimates within ±45.5 days of gold standard ultrasound/BOE dating. In statistical models that included three serial measurements of SFH during pregnancy to predict GA, the

Table 3. Studies reporting upon the accuracy of symphysis fundal height to estimate gestational age (higher quality studies with ultrasound-confirmed dating).

| Author | Year | Study Setting (NICU/clinic/hospital/community, district/city, country) | Sample Size | SFH Measurement/conversion to GA | Correlation (R) with reference GA | Mean difference/bias (days) (SFH—reference GA) | SD of the mean GA difference (days) | Bland Altman 95% LOA (LL, UL) [days] | 95% CI prediction error | % within 7 days | % within 14 days | Sensitivity (%) | Specificity (%) | PPV (%) | NPV (%) |
|---|---|---|---|---|---|---|---|---|---|---|---|---|---|---|---|
| Althabe | 2015 | 1) Argentina; 2) India 3) Pakistan; 4) Zambia | 1029 | Color coded tape | – | – | – | – | – | – | – | <36wk 1)87; 2)78 3) 63; 4)91 | <36wk 1)51; 2)89 3) 94; 4)50 | – | – |
| White | 2012 | Antenatal clinic, Thai-Burmese border | 2437 | Statistical model, 3 measures | – | – | 16.6* | – | (–36, 29) | – | 62 | – | – | – | – |
| van Rensburg | 2003 | Primary health center, Bloemfontein, South Africa | 173 | SFH cm = GA wk | – | –11.2 | 17.6* | (–23.4, 45.8) | – | – | 59 | – | – | – | – |
| Karl | 2015 | Primary health centers, Madang, PNG | 688 | Linear-White model | 0.49 | 0 | – | (–26, 26) | – | – | – | 72 | 87 | 23 | 98 |
| | | | 502 | Sequential-White model | 0.21 | 4 | 11.5* | (–19, 26) | – | – | – | 43 | 96 | 40 | 97 |
| Malaba | 2018 | Primary center, S Africa | 261 | NS | – | – | – | – | – | 36% concordance[1] | – | – | – | – | – |
| Moore | 2015 | Clinic, Thai-Myanmar | 704 | White model, 3 measures | – | 1.12 | 7.42 | – | – | – | – | 21 | 99 | – | – |
| Jehan | 2010 | Community-based, Hyderabad, Pakistan | 1128 | SFH cm = GA wk | – | 3.08 | 11.9 | – | – | 75 | 91 | 67.8 | 95.8 | 77.6 | 93.3 |
| Lee | 2020 | Community-based, Sylhet district, Bangladesh | 1486 | SFH cm = GA wk | | –30.8 | 28.2* | (–87, 26) | | | | <34wk 83; | <34wk 71; | <34wk 80.8; | <34wk 73.2; |
| | | | | Model, 1 measure | 0.70 | – | 26.9* | – | +/– 53.3 | – | 40 | <37wk | <37wk | <37wk | <37wk |
| | | | | Model, 3 measures | 0.71 | – | 25.9* | – | +/– 51.7 | – | 69 | 81 | 67 | 88.2 | 53.7 |
| van Bogaert | 1999 | Tertiary hospital, Eastern Cape Province, S. Africa | 800 | NS | 0.91 | – | – | – | – | – | – | – | – | – | – |

*(Continued)*

PLOS ONE | https://doi.org/10.1371/journal.pone.0272718 August 25, 2022 9 / 19

**Table 3.** (Continued)

| Author | Year | Study Setting (NICU/clinic/ hospital/ community, district/city, country) | Sample Size | SFH Measurement/ conversion to GA | GA estimated by SFH versus BOE or LMP | | | | | | | Validity to identify preterm GA (<37 weeks unless otherwise noted) | | | |
|---|---|---|---|---|---|---|---|---|---|---|---|---|---|---|---|
| | | | | | Correlation (R) with reference GA | Mean difference/ bias (days) (SFH— reference GA) | SD of the mean GA difference (days) | Bland Altman 95% LOA (LL, UL) [days] | 95% CI prediction error | % within 7 days | % within 14 days | Sensitivity (%) | Specificity (%) | PPV (%) | NPV (%) |
| Shrestha | 2017 | Banke District, Nepal | 614 | NS | 0.40 | – | – | – | – | 19 | 62 | – | – | – | – |

(–) indicates that the data was not available for that paper

*Numbers were calculated by authors of this paper

[1]Concordance defined by American College of Obstetricians and Gynecologists: <7 days between 14–15 weeks, <10 days between 16–21 weeks, and <14 days between 22–27 weeks

Abbreviations: SFH = symphysis fundal height, NS = not stated, GA = gestational age, AGA = appropriate-size-for-gestational age, SD = standard deviation, LOA = limits of agreement, LL = lower limit, UL = upper limit, CI = confidence interval, PPV = positive predictive value, NPV = negative predictive value, BOE = BOE, LMP = last menstrual period

precision error was improved. Based on four studies (n = 4391 pregnancies), GA estimation using three serial SFH measurements had a mean bias of -1.9 days and pooled SD of 17.1 days, with a 95% prediction window for GA of ±33.4 days [44, 54, 66, 68]. In sensitivity analysis excluding the Bangladesh study [44], the pooled SD was 14.6 days, with a 95% prediction window of ±29.6 days.

Several studies reported the percentage of pregnancies that would be dated by SFH within ±7 days or ±14 days of ultrasound dating. Compared to ultrasound-confirmed GA, SFH-based dates were within ±7 days for 19% [69] to 75% [45] of pregnancies in two studies. The percentage of pregnancies that would be dated by SFH within ±14 days of ultrasound dating ranged from 40% to 91% in five studies [44, 45, 54, 69, 70]. In a pooled analysis of these five studies (n = 5838 pregnancies), an estimated 71% (95% CI: 66–77%) of pregnancies were dated within ±14 days of ultrasound confirmed dating.

Nine studies reported Pearson correlation coefficients for GA estimated by SFH compared to GA estimated by a gold standard technique. Compared to ultrasound, correlation coefficients ranged from 0.21 to 0.91 (median 0.55; n = 4 studies) [44, 53, 66, 69].

### Accuracy of SFH to identify preterm gestational age

Several studies in this review had also assessed the diagnostic accuracy of SFH to identify preterm GA at varying thresholds of GA (Table 3). Data were not pooled because of the wide range of thresholds for preterm GA as well as cut-off values of SFH to identify preterm GA.

In the multi-country Antenatal Corticosteroids Trial (ACT), the National Institute of Child Health and Human Development (NICHD) Global Network used a cut-off of the mean 10th percentile of SFH measures at the 36th week of completed gestation based on regional population-based data to estimate the sensitivity and specificity of the SFH cut-off to classify prematurity (<36 weeks). The cut-off performed better in India (78% sensitivity, 89% specificity) and Pakistan (63% sensitivity, 94% specificity) than in Argentina (87% sensitivity, 51% specificity) and Zambia (91% sensitivity, 50% specificity) [76].

Four studies assessed the validity of SFH cut-offs to identify prematurity <37 weeks. In one study, using the 1cm = 1wk GA clinical assumption had a sensitivity of 68% and a specificity of 96% [45]. Karl et al. calculated the sensitivity and specificity of statistical models using a single SFH measure as well as three serial SFH measures to predict preterm GA (<37 weeks) [66]. A single SFH measurement had a sensitivity of 72% and specificity of 87% to predict preterm birth, while three serial SFH measurements had 43% sensitivity and 96% specificity. Similar to Karl et al., Moore et al. reported low sensitivity (21%) and high specificity (99%) for preterm classification by three serial SFH measurements compared to early ultrasound, with misclassification of 79% of preterm newborns as term [68]. Lee et al. reported that an SFH cut-off of <30 cm had 81% sensitivity and 67% specificity for classification of preterm GA <37 weeks, and that an SFH cut-off of <29 cm had 83% sensitivity and 71% specificity for classification of preterm GA <34 weeks, which is the threshold for providing antenatal corticosteroids [44].

### Inter- and intra-rater reliability of SFH measurements

Ten studies had at least 50 subjects for inter-rater and/or intra-rater reliability assessments (S4 Table in S1 Appendix). For inter-rater reliability, the mean difference between measurers was reported to be 0.66 cm (SD ±1 cm) [77], 0.88 cm (95% LOA ±3.65 cm) [44], and 2.06 cm [78]. Rogers et al. and Lee et al. reported that 95% and 70% of measurements, respectively, between two measurers were within 2 cm of each other, and Althabe et al. reported that 95% of measurements between two measurers were within 2–3 cm [44, 47, 77]. For intra-rater reliability, Papageorghiou et al. reported that the mean difference between two SFH measurements by the

same measurer was 0.07 cm with 95% LOA of 1.5 cm [2], and Engstrom *et al.* reported a mean difference of 1.13 cm [78] between measurements by the same measurer.

## Discussion

### Main findings

In our systematic review, we identified a wide range of techniques to measure SFH (Table 1) and methods by which to calculate GA from SFH measurements, and highly variable inter-rater reproducibility of measurements (S2 Table in S1 Appendix). Based on our pooled analysis, the common clinical assumption that 1 cm SFH is equal to 1 week of gestation dated pregnancies with a wide margin of error of ±43 days compared to ultrasound or best obstetric estimate based dating. Statistical models using three serial SFH measurements performed somewhat better, dating 95% of pregnancies within ±33 days of ultrasound or BOE based GA estimates.

### Strengths & limitations

To our knowledge, this is one of the few reports to systematically examine and report upon the use of SFH for the purpose of GA dating (as opposed to monitoring or identification of fetal growth restriction), and specifically in LMIC settings where SFH is used for this purpose given limited access to more accurate GA methods like ultrasound. It is in these very settings where rates of pregnancy morbidity and fetal growth restriction are high, and thus SFH to GA conversion methods based on highly-selective study populations and optimal fetal growth would systematically underestimate many pregnancies in LMIC. We extensively searched a range of databases and gray literature to identify potential data sources. A wide range of measurement techniques are summarized, as well as conversion methods to translate an SFH measurement to a GA. We describe 37 studies across 5 regions, with 33,346 study participants, and summarize 19 different measuring techniques and 12 different ways clinicians/ researchers have converted SFH into GA.

The study quality for a majority of studies was low, as many did not clearly describe the technique of SFH measurement or the method used to calculate gestational age from SFH. Forty-one percent of studies did not have a high-quality gold standard dating method (ultrasound or best obstetric estimate including ultrasound). Additionally, while a common clinical algorithm is the use of a combination of measures to estimate GA, such as SFH in combination with LMP, we did not have adequate data to assess the accuracy of combined methods. Data availability within the parameters of our inclusion criteria was a limitation for analysis. Additionally, although the studies included in our analyses had ultrasound-confirmed dates as the reference standard, different SFH measuring techniques, SFH to GA conversion methods, and studying sampling methods were used.

### Interpretation in light of other evidence

The accuracy of using SFH to estimate GA has long been debated, however SFH continues to be used in LMIC for GA dating due to the lack of other routinely available and feasible methods for GA determination. Based on our findings, SFH should not be used as the sole method of determining GA because SFH estimates of GA were quite inaccurate compared to ultrasound confirmed dating, with a wide margin of error.

The recent publication of the results of the WHO Antenatal Corticosteroids for Improving Outcomes in preterm Newborns (ACTION) Trial underscores the importance of using accurate GA dating methods like ultrasound instead of SFH for clinical decision-making to manage

preterm birth [79]. The previous Global Network's Antenatal Corticosteroids Trial—which used LMP or SFH as the basis for identifying women at risk of preterm labor for targeted administration of dexamethasone—failed to show benefits among small infants (<5[th] percentile for birth weight, the trial's proxy for preterm) and was associated with an overall increase in neonatal mortality, stillbirth, and suspected maternal infection in the intervention group [80]. The inaccuracy of GA determination by LMP and SFH was considered to be a potential reason for these unexpected findings as there may have been inclusion of term but growth-restricted fetuses in the trial [81]. As a result of the Global Network Trial, in 2015 the WHO recommended that antenatal corticosteroids be used only under certain conditions, including the accurate assessment of GA by early ultrasound. When ultrasound was used to determine GA and identify women at risk of preterm birth in the WHO ACTION Trial for corticosteroid treatment, dexamethasone was associated with significant reductions in neonatal death, and stillbirth compared to placebo [79].

Therefore, our recommendations are to minimize or even eliminate the role of SFH in GA estimation during clinical antenatal care and research, and to instead prioritize increasing coverage of early ultrasound as well as training in ultrasonography in LMIC. Expanding access to ultrasound would ensure that pregnancy monitoring and delivery of life-saving interventions like antenatal corticosteroids are guided by accurate estimation of gestational age.

Several important studies reported data on the relationship between SFH and gestational age but were excluded from the review but are detailed in supplementary materials. (S1 Table in S1 Appendix) Because these study cohorts were aimed to develop standards for optimal fetal growth and were restricted to healthy, low-risk women without pregnancy morbidities; the relationships between GA and SFH in these studies would not be generalizable to general obstetric populations in LMICs. With the aim of creating global prescriptive SFH "standards" for fetal growth monitoring, the INTERGROWTH-21[st] study, and growth standards for SFH [2], applied rigorous selection criteria that excluded pregnancies with known socioeconomic and health constraints on fetal growth. The study excluded women of low and high BMI and women with significant comorbidities who comprise a large proportion of pregnancies in LMICs. In an indigenous pregnant population in Panama [9] with high rates of UTI, hookworm, and undernutrition, the prevalence of SFH <10% according to the INTER-GROWTH standard was 50.6%, compared to 8% using the local SFH reference. If the gestational age is known, use of INTERGROWTH-21[st] SFH charts would flag pregnancies with suspected fetal growth restriction in these settings. However, if accurate GA is not known and SFH is used to establish GA rather than screen for growth abnormalities, use of INTER-GROWTH-21[st] SFH standards would systematically underestimate GA when applied to general obstetric populations such as this or in Asia, where fetal growth restriction or pregnancy morbidities are prevalent [2, 8]. SFH measurement appears to be more accurate for GA estimation when the study populations are selected for optimal fetal growth as the assumption that size is equivalent to age is more likely to be valid. For example, the agreement between SFH and ultrasound-based GA was relatively higher in the NICHD Fetal Growth Study, that enrolled only pregnant women with pregravid BMI 19–29.9 kg/m2, without pre-existing medical diagnoses or pregnancy diseases (gestational diabetes, pre-eclampsia) [1]. However, we excluded such studies from our analysis on accuracy of SFH for GA estimation to reflect real-world conditions and have highlighted how poor SFH performs in the general obstetric population in LMICs.

The variation in the definition of SFH and how it is measured contributes to a body of literature in which researchers and practitioners are measuring different things, in different ways, and interpreting their measurements differently. We identified 19 different SFH measurement techniques (Table 1) and 12 different ways in which clinicians/researchers have converted

SFH to gestational age (S2 Table in S1 Appendix), ranging from simple rules like McDonald's "1 to 1" rule to complex regression equations. Inter-rater reproducibility of measurements is also highly variable across studies. Many dynamic factors increase variability, including transverse and oblique fetal lies, Braxton Hicks contractions, the fullness of the woman's bladder, and fetal movements [82]. It is therefore unsurprising that, based on our systematic review, the limits of agreement for SFH are wide compared to gold standard GA estimates and that inter- and intra-observation variation is high. Other factors may influence SFH measurement that we were not able to address in this review, including maternal obesity [1, 67, 73] and timing in pregnancy of SFH measurements [1].

## Conclusion

Accurate estimates of GA are needed for safe and effective antenatal care, including fetal growth monitoring and decision-making about interventions for high-risk pregnancies. Our systematic review and meta-analysis assessing the agreement between SFH and ultrasound-confirmed pregnancy dating found that SFH had wide margins of error, which we feel are unacceptably wide to be clinically relevant. Furthermore, there needs to be greater awareness of the existing variability in SFH definitions, measuring techniques, and conversions, which further comprises clinical utility for GA dating. Even though it is often the only method used or available, SFH measurement has low accuracy and may not be possible to improve. This study underscores the importance of improving coverage of early pregnancy ultrasound scans and new ultrasound techniques to improve GA assessment in late pregnancy.

## Supporting information

**S1 Checklist. PRISMA 2009 checklist.**
(DOCX)

**S1 Appendix.**
(DOCX)

## Acknowledgments

Merab Nnyishime, Dr. Hema Magge, and Dr. Dilys Walker for sharing clinical standards for SFH.

## Author Contributions

**Conceptualization:** Rachel Whelan, Lauren Schaeffer, Blair J. Wylie, Anne C. C. Lee.

**Data curation:** Rachel Whelan, Lauren Schaeffer, Ingrid Olson, Lian V. Folger, Saima Alam, Nayab Ajaz, Karima Ladhani.

**Formal analysis:** Rachel Whelan, Lauren Schaeffer, Ingrid Olson, Bernard Rosner, Anne C. C. Lee.

**Funding acquisition:** Anne C. C. Lee.

**Methodology:** Rachel Whelan, Lauren Schaeffer, Lian V. Folger, Blair J. Wylie, Anne C. C. Lee.

**Project administration:** Ingrid Olson.

**Writing – original draft:** Rachel Whelan, Lauren Schaeffer.

**Writing – review & editing:** Ingrid Olson, Lian V. Folger, Saima Alam, Nayab Ajaz, Karima Ladhani, Bernard Rosner, Blair J. Wylie, Anne C. C. Lee.

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
