## [Decision Letter · Decision Letter 0]

11 Apr 2022

PONE-D-21-37354Measurement of symphysis fundal height for gestational age estimation in Low-to-Middle-Income Countries: A systematic review and meta-analysisPLOS ONE

Dear Dr. Lee,

Thank you for submitting your manuscript to PLOS ONE. After careful consideration, we feel that it has merit but does not fully meet PLOS ONE’s publication criteria as it currently stands. Therefore, we invite you to submit a revised version of the manuscript that addresses the points raised during the review process.

The manuscript and the reviewers’ comments were carefully evaluated. Although the manuscript received two positive comments, the third Reviewer suggested major revisions and highlighted study limits that should be appropriately discussed. I would recommend improving the manuscript by addressing as much as possible the recommended revisions by Reviewer 3. 

We look forward to receiving your revised manuscript.

Kind regards,

Simone Garzon

Academic Editor

PLOS ONE

Journal Requirements:

3. Thank you for stating the following in the Acknowledgments/ Funding Section of your manuscript: 

The Bill and Melinda Gates Foundation.

This work was supported by the Bill & Melinda Gates Foundation through grant OPP1130198. This work was conducted with support from Harvard Catalyst | The

Harvard Clinical and Translational Science Center (National Center for Advancing

Translational Sciences, National Institutes of Health Award UL 1TR002541) and

financial contributions from Harvard University and its affiliated academic healthcare

centers. ACL was supported by a grant from the Eunice Kennedy Shriver National

Institute of Health and Child Development (K23 HD091390-01). The content is solely

the responsibility of the authors and does not necessarily represent the official views of

Harvard Catalyst, Harvard University and its affiliated academic healthcare centers, or

the National Institutes of Health. The funders had no role in study design, data

collection and analysis, decision to publish, or preparation of the manuscript.

This research was supported by a grant from the Bill and Melinda Gates Foundation

(BMGF). ACL reported research grants from the Eunice Kennedy Shriver National

Institute of Health and Child Development and the BMGF, and is a consultant to the

World Health Organization. BW and BR reported research grants from the NIH. BW

serves on the Board for the Society of Maternal-Fetal Medicine.

5. Please note that in order to use the direct billing option the corresponding author must be affiliated with the chosen institute. Please either amend your manuscript to change the affiliation or corresponding author, or email us at plosone@plos.org with a request to remove this option.

7. Your ethics statement should only appear in the Methods section of your manuscript. If your ethics statement is written in any section besides the Methods, please move it to the Methods section and delete it from any other section. Please ensure that your ethics statement is included in your manuscript, as the ethics statement entered into the online submission form will not be published alongside your manuscript. 

8. We note that you have referenced (ie. Bewick et al. [5]) which has currently not yet been accepted for publication. Please remove this from your References and amend this to state in the body of your manuscript: (ie “Bewick et al. [Unpublished]”) as detailed online in our guide for authors

Reviewers' comments:

Reviewer's Responses to Questions

**Comments to the Author**

1. Is the manuscript technically sound, and do the data support the conclusions?

Reviewer #1: Yes

Reviewer #2: Yes

Reviewer #3: Partly

2. Has the statistical analysis been performed appropriately and rigorously? 

Reviewer #1: Yes

Reviewer #2: Yes

Reviewer #3: No

3. Have the authors made all data underlying the findings in their manuscript fully available?

Reviewer #1: Yes

Reviewer #2: Yes

Reviewer #3: Yes

4. Is the manuscript presented in an intelligible fashion and written in standard English?

Reviewer #1: Yes

Reviewer #2: Yes

Reviewer #3: Yes

5. Review Comments to the Author

Reviewer #1: The literature review is accurate with the according references.

The introduction is excellent.

The methodology is clean. The SR were previously registered and the appropriate guidelines were used.

The objectives were clear. The assessment of biases and the analyses (random effect) are correct.

The conclusion and the discussion are accurate and interesting.

The importance of the paper is high.

Reviewer #2: The authors are to be commended for a well-designed study and exceptionally well-presented paper.

The data, clearly presented, conveys the limitations of utilizing symphysis fundal height measurements as an accurate estimate of gestational age.

The tables that were provided may serve as an superb guide to researchers where dating of pregnancy is key and ultrasound not available.

Regional differences in sensitivity and specificity of SFH have previously been reported; the weight of evidence presented confirms that such differences are real.

A 6-week window of variability for accurate dating is suggested, generally well in excess of the time required to assess critical perinatal outcomes.

I have no suggested edits. This is an excellent and well-referenced paper that warrants publication.

Reviewer #3: Thank you for asking me to review this interesting article on measurement of symphysis fundal height for gestational age estimation

Measuring SFH is used as a first level screening test to identify fetuses with growth aberrations. In resource poor settings SFH is used (in women where LMP is not known and where ultrasound is not available) to estimate GA.

I have a number of comments:

1. As the authors outline, ultrasound is most accurate for GA assessment. But, when it comes to GA estimation based on measurement (whether ultrasound or SFH) there is an underlying fundamental flawed assumption: namely that every fetus of a certain measurement is of the same GA. This is clearly untrue, due to normal biological variation; measurement variability; the fact that growth restriction (or fetal overgrowth) will automatically result in an under-(or over-) estimation of GA, respectively. This findamental limitation needs to come through better, as does the fact that such growth aberrations, in particular poor fetal growth, are most common in underserved regions.

2. Originality: As the authors state, previous systematic reviews have assessed SFH measurement for fetal growth monitoring. There is a recent SR on SFH for GA estimation (Second and third trimester estimation of gestational age using ultrasound or maternal symphysis-fundal height measurements: A systematic review. Self et al. doi.org/10.1111/1471-0528.17123). That study also looks at ultrasound parameters and not restricted to LMIC settings. Therefore I think the current study is still of value, but should reference the study by Self et al.

3. The lack of a unified or strong method as the comparator for GA is an important limitation in my view. As the authors themselves state, GA is best determined by ultrasound. Therefore comparing SFH to BOE or LMP is of very limited value – we do not know what the actual GA is. Please discuss this limitation and consider restricting analysis or sensitivity analysis including only ultrasound based "actual" GA.

4. The rationale is unclear as to why studies from LMIC were only included. You state this was done “because SFH charts are more likely to be used for GA dating in LMIC settings, while in HIC, pregnancies are typically dated by ultrasound and SFH charts are used for fetal growth monitoring purposes”. Charts of SFH may be constructed for dating or growth monitoring (or both), but to me there is actually an advantage of assessing HIC charts – exactly because dating is “secure”. The best strategy would be accurately dated pregnancies with SFH from LMIC settings, which is provided by Intergrowth. Please justify better your decision or better still include sich studies.

5. I question the exclusion of studies including only healthy populations – why was this done? Surely you cannot replicate SFH studies everywhere? If you include pathological pregnancies or higher BMI variability, how does that increase external validity - it will always depend on the proportion of pathology in each sub-population; and the distribution of BMI in each subpopulation. And even then you will only achieve accuracy at population level, even with "customisation". You need to explain this difference in the paper.

6. A sophisticated analysis was undertaking grouping world regions, gold standard and “a priori differences in fetal size and rates of growth restriction between Africa and Asia"…. and the authors developed separate SFH models for Africa and Asia. To me this is an over-sophisticated analysis given the many limitations of the constituent studies. It ignores or at best is hampered by the fact that the methods in different contributing populations are different… including of measurement strategy, dating etc. Differences due to region could be due to many other differences within each region such as data collection, quality, bias in measurement, etc etc. You try to address this by ensuring “only ultrasound/BOE studies were included”. I think it would be better to take the few best studies and assess this, including only the ones with best dating strategy based on early ultrasound. I would suggest revising the analysis strategy, aldo given the poont 8 below.

7. Point 6 is particularly relevant when you compare to the INTERGROWTH-21st SFH - in African LMIC settings these were similar but in Asian LMIC settings they were lower than the African cohorts. It should be explored what are the reasons for this in particular as the data were driven by a single study from Bangladesh.

8. Finally, to me the biggest issues are conceptual:

You show that prescriptive SFH standards suggest a high prevalence of SFH <10% in resource poor settings - poor settings have high levels of FGR. This also means that using SFH would underestimate GA in these same populations where FGR or pregnancy morbidities are prevalent. This however goes back to my point (1) in this review: you cannot know in an individual woman whether growth restriction is present or whether the comorbidity has affected fetal growth. Hence, I fundamentally disagree with your assertion that “the proposed regional population-based reference SFH values are clinically relevant and important for GA estimation in LMIC settings” and that “regional population-based reference SFH charts or customized SFH charts may provide a relatively better estimate of GA than international SFH standards”. Such an approach may better estimate GA at POPULATION level, but it is NOT useful clinically for the care of INDIVIDUAL women because you do not know the fetal size in the particular mother! You rightly caution, that “GA estimates should still be interpreted with a prediction window of ±6 weeks”. But in truth, this renders such a method pretty ineffective: Such accuracy can be achieved at clinic level by finding the most common GA women attend (eg if most women first attend at 24 weeks you can simply label each woman as being 24 weeks pregnant, this will work similarly). If it is a really bad method then say so, do not give clinicians the poor excluse to continue with poor practice!

You cannot have it both ways: on the one hand saying: “look our methods of regional charts are very good” and on the other hand saying “but actually all methods are bad”. Nail your colors to the mast: in my view the "mast" is the right one in the conclusion of your abstract, ie “SFH is inaccurate for estimating GA and should not be used for GA dating” – leave it there! “...whenever possible” is superfluous. What you have really shown is that SFH for GA estimation is inadequate care for vulnerable populations of pregnant women and should be abandoned. Only with such uncontroversial strong recommendations will we make progress… leaving a window open will always result in inadequate public health investment: the narrative will be “no we do not need ultrasound, CC Lee says we need more relevant SFH charts”, neglecting the +/- 6 week accuracy.

Do not undesetimate your power to shape policy on the ground, make the right decision! I would advise to change the narrative.

Minor comment:

I was unable to locate ref 2 “Intergrowth. INTERGROWTH-21st International Fetal and Newborn Growth Standards for the 21st Century”. Perhaps this should be the relevant Intergrowth website, if so please add this and the access date, or is it a paper you are referring to?

6. PLOS authors have the option to publish the peer review history of their article (what does this mean?). If published, this will include your full peer review and any attached files.

Reviewer #1: No

Reviewer #2: No

Reviewer #3: No

---

## [Author Response · Author response to Decision Letter 0]

26 May 2022

Response to Reviewers

PONE-D-21-37354

Measurement of symphysis fundal height for gestational age estimation in Low-to-Middle-Income Countries: A systematic review and meta-analysis

Journal Requirements:

Author Response: We have checked and updated the format for consistency with Plos One.

Author Response: We have included the correct funding statement in the cover letter.

3. Thank you for stating the following in the Acknowledgments/ Funding Section of your manuscript: 

The Bill and Melinda Gates Foundation.

This work was supported by the Bill & Melinda Gates Foundation through grant OPP1130198. This work was conducted with support from Harvard Catalyst | The

Harvard Clinical and Translational Science Center (National Center for Advancing

Translational Sciences, National Institutes of Health Award UL 1TR002541) and

financial contributions from Harvard University and its affiliated academic healthcare

centers. ACL was supported by a grant from the Eunice Kennedy Shriver National

Institute of Health and Child Development (K23 HD091390-01). The content is solely

the responsibility of the authors and does not necessarily represent the official views of

Harvard Catalyst, Harvard University and its affiliated academic healthcare centers, or

the National Institutes of Health. The funders had no role in study design, data

collection and analysis, decision to publish, or preparation of the manuscript.

Author Response: We have included the correct funding statement in the cover letter.

This research was supported by a grant from the Bill and Melinda Gates Foundation

(BMGF). ACL reported research grants from the Eunice Kennedy Shriver National

Institute of Health and Child Development and the BMGF, and is a consultant to the

World Health Organization. BW and BR reported research grants from the NIH. BW

has served on the Board for the Society of Maternal-Fetal Medicine within the past three years.

Author Response: We have amended the statement in our cover letter. 

5. Please note that in order to use the direct billing option the corresponding author must be affiliated with the chosen institute. Please either amend your manuscript to change the affiliation or corresponding author, or email us at plosone@plos.org with a request to remove this option.

Author Response: We are unclear on what exactly this means; as currently written, the corresponding author (ACL) is affiliated Brigham & Women’s Hospital, which is the institute that will be responsible for billing. However, if the “direct” billing option requires something different, we can opt out of that option. We will email the above address to clarify this and assess if action needs to be taken to change the billing option. 

Author Response: We have amended the statement in our cover letter. 

7. Your ethics statement should only appear in the Methods section of your manuscript. If your ethics statement is written in any section besides the Methods, please move it to the Methods section and delete it from any other section. Please ensure that your ethics statement is included in your manuscript, as the ethics statement entered into the online submission form will not be published alongside your manuscript. 

Author Response: We have moved the Ethics statement to the Methods section of the manuscript, and have deleted elsewhere.

8. We note that you have referenced (ie. Bewick et al. [5]) which has currently not yet been accepted for publication. Please remove this from your References and amend this to state in the body of your manuscript: (ie “Bewick et al. [Unpublished]”) as detailed online in our guide for authors

Author Response: The citation has been removed. 

Author Response: The Supporting Information file has been reformatted and renamed according to the guidelines. In-text citations have been updated accordingly. 

Reviewers' Comments to the Author

1. Is the manuscript technically sound, and do the data support the conclusions?

Reviewer #1: Yes

Reviewer #2: Yes

Reviewer #3: Partly

2. Has the statistical analysis been performed appropriately and rigorously? 

Reviewer #1: Yes

Reviewer #2: Yes

Reviewer #3: No

3. Have the authors made all data underlying the findings in their manuscript fully available?

Reviewer #1: Yes

Reviewer #2: Yes

Reviewer #3: Yes

4. Is the manuscript presented in an intelligible fashion and written in standard English?

Reviewer #1: Yes

Reviewer #2: Yes

Reviewer #3: Yes

5. Review Comments to the Author

Reviewer #1: The literature review is accurate with the according references.

The introduction is excellent.

The methodology is clean. The SR were previously registered and the appropriate guidelines were used.

The objectives were clear. The assessment of biases and the analyses (random effect) are correct.

The conclusion and the discussion are accurate and interesting.

The importance of the paper is high.

Reviewer #2: The authors are to be commended for a well-designed study and exceptionally well-presented paper.

The data, clearly presented, conveys the limitations of utilizing symphysis fundal height measurements as an accurate estimate of gestational age.

The tables that were provided may serve as an superb guide to researchers where dating of pregnancy is key and ultrasound not available.

Regional differences in sensitivity and specificity of SFH have previously been reported; the weight of evidence presented confirms that such differences are real.

A 6-week window of variability for accurate dating is suggested, generally well in excess of the time required to assess critical perinatal outcomes.

I have no suggested edits. This is an excellent and well-referenced paper that warrants publication.

Author Response: We thank the reviewers for the positive comments.

Reviewer #3: Thank you for asking me to review this interesting article on measurement of symphysis fundal height for gestational age estimation

Measuring SFH is used as a first level screening test to identify fetuses with growth aberrations. In resource poor settings SFH is used (in women where LMP is not known and where ultrasound is not available) to estimate GA.

I have a number of comments:

1. As the authors outline, ultrasound is most accurate for GA assessment. But, when it comes to GA estimation based on measurement (whether ultrasound or SFH) there is an underlying fundamental flawed assumption: namely that every fetus of a certain measurement is of the same GA. This is clearly untrue, due to normal biological variation; measurement variability; the fact that growth restriction (or fetal overgrowth) will automatically result in an under-(or over-) estimation of GA, respectively. This fundamental limitation needs to come through better, as does the fact that such growth aberrations, in particular poor fetal growth, are most common in underserved regions.

Author Response:

Thank you for your feedback and emphasis on this critical point. We have revised the Background section (page 3, paragraph 2) to clearly describe these fundamental flaws/limitations of SFH. 

 “When ultrasound and reliable LMP are not available, SFH is frequently used for GA estimation because it is a simple, low-cost, and feasible technique that can be performed by lay health workers [3]. However, fundamental flaws in using SFH for this purpose include the underlying assumption that fetal size approximates GA, and that every fetus of a certain size is the same GA. Fetal size is influenced by genetic factors and normal biologic variation, and fetal growth is influenced by maternal nutrition, health, and morbidities, including infections, pregnancy complications, or environmental exposures. Risk factors for poor fetal growth are much more prevalent in LMICs [8-10]. The use of standard SFH curves from high income settings with low prevalence of these risk factors would, thus, tend to systematically underestimate GA when applied to a population with high prevalence of fetal growth restriction. Further confounding the use of SFH is the lack of standardized methods for measurement of SFH and converting the measurement to gestational age.”

2. Originality: As the authors state, previous systematic reviews have assessed SFH measurement for fetal growth monitoring. There is a recent SR on SFH for GA estimation (Second and third trimester estimation of gestational age using ultrasound or maternal symphysis-fundal height measurements: A systematic review. Self et al. doi.org/10.1111/1471-0528.17123). That study also looks at ultrasound parameters and not restricted to LMIC settings. Therefore I think the current study is still of value, but should reference the study by Self et al.

Author Response: Thank you for bringing up this recent systematic review to our attention. This was published after our searches were completed. We have now included reference to this citation in paragraph 2 in our Introduction as well as referenced it in the discussion.

3. The lack of a unified or strong method as the comparator for GA is an important limitation in my view. As the authors themselves state, GA is best determined by ultrasound. Therefore comparing SFH to BOE or LMP is of very limited value – we do not know what the actual GA is. Please discuss this limitation and consider restricting analysis or sensitivity analysis including only ultrasound based "actual" GA.

Author Response: 

Thank you for this important feedback. In our revised submission, based on the reviewers feedback in the main manuscript, we now only present the data and analysis restricted to the high quality gold standard of ultrasound confirmed dating. Furthermore, in the studies that we had originally classified as “Best Obstetric Estimate/BOE” we went to the original papers and confirmed that gestational age was confirmed in all included studies by ultrasound alone or utilized as confirmation of menstrual dates. Those studies basing GA on menstrual dates alone without incorporation of ultrasound were considered separately with the LMP category of papers. All studies and analyses with LMP GA reference are relegated to and presented in the Supplementary Appendix.

In the Study Quality Assessment section, we have revised the statement “Studies with a gold standard GA based on ultrasound or a BOE that included ultrasound (LMP confirmed by ultrasound) were graded as highest quality” to now state “Studies with a gold standard GA based on ultrasound or by ultrasound-confirmation of the menstrual dates (hereafter referred to as ‘ultrasound-confirmed’ GA) were graded as highest quality given that the ultrasound confirmation of dating would be considered as closest to the “actual” truth.”

For clarification purposes, we have also revised our statements to emphasize that studies with ultrasound-confirmed dates were graded as highest quality. Since Best Obstetric Estimate (BOE) is a term used in clinical practice and may include pregnancies dated by LMP if ultrasound is not available, we have removed this language from the manuscript and replaced BOE with “ultrasound-confirmed dating.” This has been updated in all relevant sections of the manuscript.

4. The rationale is unclear as to why studies from LMIC were only included. You state this was done “because SFH charts are more likely to be used for GA dating in LMIC settings, while in HIC, pregnancies are typically dated by ultrasound and SFH charts are used for fetal growth monitoring purposes”. Charts of SFH may be constructed for dating or growth monitoring (or both), but to me there is actually an advantage of assessing HIC charts – exactly because dating is “secure”. The best strategy would be accurately dated pregnancies with SFH from LMIC settings, which is provided by Intergrowth. Please justify better your decision or better still include such studies.

Author Response: We apologize for the lack of clarity. Studies from all income settings, including HIC, were included in the review, including methods/techniques of measurement, accuracy, and inter-rater agreement. The only section in which we had originally limited to LMICs was for the weighted regression modeling of average SFH by region. We have now clarified this in the study methods. Based upon this reviewer’s feedback, we decided to remove that modeling (see point 8 below) and construction of charts for potential use for dating. These are now eliminated from the manuscript entirely. We have only retained the existing studies that report upon normal values of SFH by gestational age in LMICs that are restricted to only studies with high quality, “secure” ultrasound confirmed dating (revised Table 2).

5. I question the exclusion of studies including only healthy populations – why was this done? Surely you cannot replicate SFH studies everywhere? If you include pathological pregnancies or higher BMI variability, how does that increase external validity - it will always depend on the proportion of pathology in each sub-population; and the distribution of BMI in each subpopulation. And even then you will only achieve accuracy at population level, even with "customisation". You need to explain this difference in the paper.

Author Response: We excluded highly-selected cohorts from this review because these populations would not be representative of the general obstetric population. SFH measurements in these selected patients with only optimally healthy pregnancies would, on average, underestimate gestational age when applied to estimate gestational age in general obstetric populations where morbidities or fetal growth restriction are prevalent. For example, the INTERGROWTH study applied rigorous selection criteria that excluded pregnancies with socio-economic and health constraints on fetal growth, in order to demonstrate the optimal growth of fetuses. The INTERGROWTH study excluded women of low and high BMI, or with significant comorbidities; these women comprise a large proportion of pregnancies in LMICs. Including studies conducted in only select, optimally healthy pregnancies would bias the relationship between SFH size and gestational age in a general obstetric population in an LMIC and therefore we intentionally excluded these studies.

We also excluded studies from highly selected unhealthy populations, as these would also not be considered representative of the general obstetric population. These kinds of studies included exclusive cohorts who were recovering from malaria, who were HIV positive, or diabetic. 

6. A sophisticated analysis was undertaking grouping world regions, gold standard and “a priori differences in fetal size and rates of growth restriction between Africa and Asia"…. and the authors developed separate SFH models for Africa and Asia. To me this is an over-sophisticated analysis given the many limitations of the constituent studies. It ignores or at best is hampered by the fact that the methods in different contributing populations are different… including of measurement strategy, dating etc. Differences due to region could be due to many other differences within each region such as data collection, quality, bias in measurement, etc etc. You try to address this by ensuring “only ultrasound/BOE studies were included”. I think it would be better to take the few best studies and assess this, including only the ones with best dating strategy based on early ultrasound. I would suggest revising the analysis strategy, aldo given the point 8 below.

Author Response:

Thank you for your candid feedback. Based upon this feedback and point 8 below, we have decided to remove the weighted regression modeling and presentation of average regional growth curves. We agree with the conclusion that SFH should be abandoned for GA dating, and have removed the modeling and average curves to avoid any confusion in our messaging. We have only retained the summary of published data on SFH size across gestation for studies with high quality ultrasound confirmed dating (Table 2). The tables and analyses in the main paper now include only those studies with high quality ultrasound-confirmed GA. Studies with the LMP gold standard are relegated to the Supporting Information (S1 Supplementary Appendix).

7. Point 6 is particularly relevant when you compare to the INTERGROWTH-21st SFH - in African LMIC settings these were similar but in Asian LMIC settings they were lower than the African cohorts. It should be explored what are the reasons for this in particular as the data were driven by a single study from Bangladesh.

Author Response: 

Based on reviewer feedback, we removed the weighted regression curve analysis so no longer present these regional differences.

We also did conduct sensitivity analysis excluding the Bangladesh study, and present this sensitivity analysis in the results. Results, page 13, paragraphs 2 and 3.

“In sensitivity analysis excluding the Bangladesh study [44], the pooled SD was 14.6 days, with a 95% prediction window of ±29.6 days.”

8. Finally, to me the biggest issues are conceptual:

You show that prescriptive SFH standards suggest a high prevalence of SFH <10% in resource poor settings - poor settings have high levels of FGR. This also means that using SFH would underestimate GA in these same populations where FGR or pregnancy morbidities are prevalent. This however goes back to my point (1) in this review: you cannot know in an individual woman whether growth restriction is present or whether the comorbidity has affected fetal growth. Hence, I fundamentally disagree with your assertion that “the proposed regional population-based reference SFH values are clinically relevant and important for GA estimation in LMIC settings” and that “regional population-based reference SFH charts or customized SFH charts may provide a relatively better estimate of GA than international SFH standards”. Such an approach may better estimate GA at POPULATION level, but it is NOT useful clinically for the care of INDIVIDUAL women because you do not know the fetal size in the particular mother! You rightly caution, that “GA estimates should still be interpreted with a prediction window of ±6 weeks”. But in truth, this renders such a method pretty ineffective: Such accuracy can be achieved at clinic level by finding the most common GA women attend (eg if most women first attend at 24 weeks you can simply label each woman as being 24 weeks pregnant, this will work similarly). If it is a really bad method then say so, do not give clinicians the poor excluse to continue with poor practice!

You cannot have it both ways: on the one hand saying: “look our methods of regional charts are very good” and on the other hand saying “but actually all methods are bad”. Nail your colors to the mast: in my view the "mast" is the right one in the conclusion of your abstract, ie “SFH is inaccurate for estimating GA and should not be used for GA dating” – leave it there! “...whenever possible” is superfluous. What you have really shown is that SFH for GA estimation is inadequate care for vulnerable populations of pregnant women and should be abandoned. Only with such uncontroversial strong recommendations will we make progress… leaving a window open will always result in inadequate public health investment: the narrative will be “no we do not need ultrasound, CC Lee says we need more relevant SFH charts”, neglecting the +/- 6 week accuracy.

Do not undesetimate your power to shape policy on the ground, make the right decision! I would advise to change the narrative.

Author Response: We agree with the reviewer, that SFH should not be used for GA dating, particularly in vulnerable pregnant populations. Thus to clarify our messaging and based on recommendations of the reviewer, we have removed the weighted regression modeling and average regional SFH curves from the manuscript methods and results. We have also clarified the messaging in the Discussion and Conclusions. 

For example, in the Conclusions section, we state:

“Our systematic review and meta analysis assessing the agreement of SFH and gold standard pregnancy dating found that SFH had wide margins of error, which we feel are unacceptably wide to be clinically relevant….This study underscores the importance of improving coverage of early pregnancy ultrasound scans and new ultrasound techniques to improve GA assessment in late pregnancy.” 

Minor comment:

I was unable to locate ref 2 “Intergrowth. INTERGROWTH-21st International Fetal and Newborn Growth Standards for the 21st Century”. Perhaps this should be the relevant Intergrowth website, if so please add this and the access date, or is it a paper you are referring to?

Author Response:

We have corrected citation 2 to:

Papageorghiou AT, Ohuma EO, Gravett MG, Hirst J, da Silveira MF, Lambert A, et al. International standards for symphysis-fundal height based on serial measurements from the Fetal Growth Longitudinal Study of the INTERGROWTH-21st Project: prospective cohort study in eight countries. BMJ (Clinical research ed). 2016;355. doi: 10.1136/bmj.i5662.

6. PLOS authors have the option to publish the peer review history of their article (what does this mean?). If published, this will include your full peer review and any attached files.

Do you want your identity to be public for this peer review? For information about this choice, including consent withdrawal, please see our Privacy Policy.

Reviewer #1: No

Reviewer #2: No

Reviewer #3: No

---

## [Decision Letter · Decision Letter 1]

26 Jul 2022

Measurement of symphysis fundal height for gestational age estimation in low-to-middle-income countries: A systematic review and meta-analysis

PONE-D-21-37354R1

Dear Dr. Lee,

We’re pleased to inform you that your manuscript has been judged scientifically suitable for publication and will be formally accepted for publication once it meets all outstanding technical requirements.

Kind regards,

Simone Garzon

Academic Editor

PLOS ONE

Additional Editor Comments (optional):

Reviewers' comments:

Reviewer's Responses to Questions

**Comments to the Author**

1. If the authors have adequately addressed your comments raised in a previous round of review and you feel that this manuscript is now acceptable for publication, you may indicate that here to bypass the “Comments to the Author” section, enter your conflict of interest statement in the “Confidential to Editor” section, and submit your "Accept" recommendation.

Reviewer #1: All comments have been addressed

Reviewer #2: All comments have been addressed

2. Is the manuscript technically sound, and do the data support the conclusions?

Reviewer #1: Yes

Reviewer #2: Yes

3. Has the statistical analysis been performed appropriately and rigorously? 

Reviewer #1: (No Response)

Reviewer #2: Yes

4. Have the authors made all data underlying the findings in their manuscript fully available?

Reviewer #1: Yes

Reviewer #2: Yes

5. Is the manuscript presented in an intelligible fashion and written in standard English?

Reviewer #1: Yes

Reviewer #2: Yes

6. Review Comments to the Author

Reviewer #1: The authors addressed most of the concerns.

The rules for SR and MA have been adequately followed.

The originality and interest of the paper are important.

Reviewer #2: This well written paper is a systematic review describing measurements of symphysis fundal height (SFH) for gestational age estimation in low and middle-income countries. The systematic review included 37 (of 1,003 studies identified) 30 of which provided data for Africa and South Asia.

The overall “take-away message” for this meta-analysis is that measurement of symphysis fundal height is not a good tool for estimating gestational age. The article reinforces the importance of obtaining early ultrasound for assessment, where such knowledge can impact clinical management.

Nineteen different measurement techniques were noted among the > 33,000 participants. The sensitivity and specificity of SFH was found to be lower than other commonly used assessment tools.

The “one cm – one week rule” was found not to be accurate and SFH measurements in Asians was consistently lower for the same period of gestation than among women in Africa.

While it would have been helpful to determine at what time in pregnancy such measurements had been obtained, the poor predictive value of such measurements in LMICs would suggest that these not be primarily used in the management of patients. However, data presented will be helpful in further research initiatives related to pregnancy dating.

I have no edits or corrections. This paper is worthy of publication.

7. PLOS authors have the option to publish the peer review history of their article (what does this mean?). If published, this will include your full peer review and any attached files.

Reviewer #1: No

Reviewer #2: No

---

## [Editor Report · Acceptance letter]

5 Aug 2022

PONE-D-21-37354R1 

Measurement of symphysis fundal height for gestational age estimation in low-to-middle-income countries: A systematic review and meta-analysis 

Dear Dr. Lee:

I'm pleased to inform you that your manuscript has been deemed suitable for publication in PLOS ONE. Congratulations! Your manuscript is now with our production department. 

Kind regards, 

on behalf of

Dr. Simone Garzon 

Academic Editor

PLOS ONE